# Evaluation of Vaccine Strategies among Healthcare Workers during COVID-19 Omicron Outbreak in Taiwan

**DOI:** 10.3390/vaccines12091057

**Published:** 2024-09-17

**Authors:** Min-Ru Lin, Chung-Guei Huang, Cheng-Hsun Chiu, Chih-Jung Chen

**Affiliations:** 1Division of Pediatric Infectious Diseases, Departments of Pediatrics, Chang Gung Memorial Hospital, Taoyuan 333, Taiwan; 2School of Medicine, College of Medicine, Chang Gung University, Taoyuan 333, Taiwan; 3Department of Laboratory Medicine, Chang Gung Memorial Hospital, Linkou Branch, Taoyuan 333, Taiwan; 4Department of Medical Biotechnology and Laboratory Science, College of Medicine, Chang Gung University, Taoyuan 333, Taiwan; 5Molecular Infectious Diseases Research Center, Chang Gung Memorial Hospital, Taoyuan 333, Taiwan

**Keywords:** COVID-19, vaccine, reactogenicity, immunogenicity, effectiveness

## Abstract

Background/Objectives: This study aimed to assess the reactogenicity and immunogenicity of various SARS-CoV-2 vaccines and compare their protective effects against COVID-19 among healthcare workers (HCWs) during the Omicron outbreak in Taiwan. Methods: Conducted from March 2021 to July 2023, this prospective observational study included healthy HCWs without prior COVID-19 immunization. Participants chose between adenovirus-vectored (AstraZeneca), mRNA (Moderna, BioNTech-Pfizer), and protein-based (Medigen, Novavax) vaccines. Blood samples were taken at multiple points to measure neutralizing antibody (nAb) titers, and adverse events (AEs) were recorded via questionnaires. Results: Of 710 HCWs, 668 (94.1%) completed three doses, and 290 (40.8%) received a fourth dose during the Omicron outbreak. AEs were more common with AstraZeneca and Moderna vaccines, while Medigen caused fewer AEs. Initial nAb titers were highest with Moderna but waned over time regardless of the vaccine. Booster doses significantly increased nAb titers, with the highest levels observed in Moderna BA1 recipients. The fourth dose significantly reduced COVID-19 incidence, with Moderna BA1 being the most effective. Conclusions: Regular booster doses, especially with mRNA and adjuvant-protein vaccines, effectively enhance nAb levels and reduce infection rates, providing critical protection for frontline HCWs during variant outbreaks.

## 1. Introduction

Severe acute respiratory syndrome coronavirus 2 (SARS-CoV-2), which has caused a global pandemic since December 2019, presents a significant challenge to global public health [1]. As of 7 January 2024, there have been more than 774 million confirmed cases and over seven million deaths worldwide [2]. Vaccination has become a critical strategy in combating this rapidly spreading infectious threat. Approximately two years into the pandemic, several vaccines, including adenovirus vector vaccines (such as AstraZeneca/Oxford), mRNA vaccines (such as BioNTech-Pfizer and Moderna), and adjuvant-protein vaccines (like Novavax and Medigen), have been developed and distributed globally [3,4,5,6,7]. These vaccines have shown effectiveness in protecting against COVID-19. However, there are new challenges for the long-term management of the emerging disease, including diminishing immunity observed six months after completing immunization [8,9], the rise of viral variants with increased transmissibility and the ability to evade immune responses [10,11], and potentially severe adverse events associated with vaccines [12,13].

Attributed to Taiwan’s effective pandemic prevention policies, the country with 23 million population had recorded only 14,603 local COVID-19 cases by the end of 2021. These cases primarily stemmed from the first community outbreak, which occurred from May to August 2021, predominantly involving the Alpha (B.1.1.7) and Beta (B.1.351) variants (Figure 1). The vaccination campaign in Taiwan began with the introduction of the AstraZeneca adenoviral vector vaccine in March 2021, followed by the Moderna mRNA vaccine in June 2021, the Medigen protein-based subunit vaccine in July 2021, and the Pfizer-BioNTech mRNA vaccine in September 2021. Despite the majority of the population having completed their primary vaccination series (two doses), and with an additional booster dose (three doses) administered to healthcare workers (HCWs) and individuals older than 65 years following recommendations from the Taiwan Centers for Disease Control (Taiwan CDC), Taiwan encountered a second significant community outbreak starting in early April 2022. This outbreak was dominated by the Omicron (B.1.1.529) variant. The Omicron variant has demonstrated a heightened ability to evade infection- and vaccine-induced neutralizing antibodies due to its mutations associated with the receptor-binding domain (RBD) and the N-terminal domain (NTD) of spike proteins [14,15]. Furthermore, studies have shown a decreased vaccine effectiveness against symptomatic infection by the Omicron variant following the completion of the two-dose primary series of any vaccine platform [16,17].

The global fight against COVID-19 requires ongoing evaluation of vaccine strategies, particularly for frontline HCWs at increased risk. Despite initial vaccine efficacy, the emergence of variants like Omicron and varied reactogenicity necessitate assessing different vaccine platforms and schedules. Prior studies show declining antibody titers over time, underscoring the need for booster doses. However, comprehensive data on the immunogenicity and reactogenicity of mix-and-match (heterologous) vaccination schedules and the comparative effectiveness of different vaccines are lacking, particularly after multiple doses. This study filled these gaps by evaluating immune responses, documenting adverse events, and assessing the protective effects of various immunization strategies against the Omicron variant. Our findings will inform public health policies and optimize vaccination strategies to enhance protection for HCWs and the broader population against current and future SARS-CoV-2 variants.

## 2. Materials and Methods

### 2.1. Ethics Statement

The study protocol was reviewed and approved by the Research Ethics Committee of the Chang Gung Memorial Hospital in 2021 (No. 202100410B0C602). Written informed consent was obtained from participants upon enrollment into this study. All experiments in this study were performed in accordance with relevant guidelines and regulations.

### 2.2. Study Design

Since 22 March 2021, a national immunization campaign against COVID-19 has officially started in Taiwan. HCWs were among the priority groups for immunization. The AstraZeneca adenoviral vector vaccine and the Moderna mRNA vaccine were the first ones to be introduced, followed by the Medigen protein-based subunit vaccine and the Pfizer-BioNTech mRNA vaccine in 2021. The Novavax subunit vaccine and the new generation Moderna BA1 vaccine became available in the second year (2022) of the immunization campaign. This study was mainly designed to assess the immunogenicity prospectively and the reactogenicity of different SARS-CoV-2 vaccines administered to HCWs in a single institute in northern Taiwan, whether given on a homologous or in a mix-and-match schedule. The protective effects of different vaccines and distinct vaccination schedules against COVID-19 were also evaluated retrospectively at the end of the study. 

### 2.3. Participants

The study’s participants were healthy healthcare workers who had not received the SARS-CoV-2 vaccine and had no intention of receiving immunosuppressive or immune-modifying medication. They should not have received a COVID-19 vaccination or participated in any investigational study. The main exclusion criteria included a history of laboratory-confirmed SARS-CoV-2 infection and a severe allergic reaction or anaphylaxis to any components of the vaccines.

### 2.4. Immunizations, Blood Samplings and Monitoring of Adverse Events

During the immunization campaign, HCWs were given the choice between adenovirus-vectored vaccines (AstraZeneca/Oxford, UK) and mRNA vaccines (Moderna, Cambridge, MA, USA) for the initial and first booster (first and second dose) shots. For the second booster (third dose), the options included two mRNA vaccines (BioNTech-Pfizer, Moderna, Cambridge, MA, USA) and an adjuvant-protein vaccine (Medigen, Taipei, Taiwan).

After enrolling, the participants were allowed to choose their preferred vaccine brand based on personal preference. The first booster dose was given approximately 2 months after the initial dose, and the second booster dose (dose 3) was administered around 7.5 months after the initial dose. The schedule followed the recommendations of the Taiwan CDC. However, strict adherence was not observed by all participants. Indeed, the average interval between the initial and second doses was 65.8 days ± 16.2 days (range, 30–165 days), and that between the initial and third doses was 226.9 days ± 21.7 days (range, 137–338 days). In December 2021, Taiwan started encouraging and providing a fourth dose of the vaccination due to the emergence and global spread of the Omicron variant. However, most HCWs in this study only received the fourth dose between May 2022 and October 2022 when the Omicron outbreak occurred.

Blood samples were collected at several time points across the immunization program: upon the first immunization (referred to as ‘Pre V1’), upon and 30 days after the second dose (‘Pre V2’ and ‘30 days post V2’), upon and 30 days after the third dose (‘Pre V3’ and ‘30 days post V3’), and upon and 30 days after the fourth dose (‘Pre V4’ and ‘30 days post V4’) of the vaccination. These samples were used for evaluating the immune response by measuring neutralizing antibody titers to SARS-CoV-2. Additionally, the detection of anti-nucleocapsid (anti-N) IgG antibody was conducted for assistance in diagnosing COVID-19. An electronic questionnaire was utilized to collect information on solicited local and systemic adverse events daily for 7 days, as well as unsolicited adverse events on a weekly basis for 28 days. At the end of the study, another questionnaire was used to gather information about COVID-19 diagnoses and symptoms that might be associated with long COVID. A confirmed COVID-19 diagnosis refers to individuals who tested positive for SARS-CoV-2 through polymerase chain reaction (PCR), antigen, or anti-N antibody testing. 

### 2.5. Measurement of Neutralizing Antibody (nAb) and Anti-N Antibody to SARS-CoV-2 

The blood samples were all evaluated using the SARS-CoV-2 antibody ELISA kit following the guidelines provided by the manufacturer, MeDiPro, based in Taiwan [18]. MeDiPro’s ELISA kit has been officially approved by the Taiwan FDA for the measurement of Spike S1- and receptor-binding domain (RBD)-binding antibodies, which serve as surrogates of live virus neutralization titers with high correlation. Values less than 12 IU/mL were considered negative results. Anti-N antibody was detected using Elecsys^®^ Anti-SARS-CoV-2 (Roche Diagnostics, Rotkreuz, Switzerland). 

### 2.6. Statistical Analysis 

The data were summarized using frequency counts and percentages for categorical data, and means and standard deviations for demographic and baseline characteristics. Categorical variables were compared using the Chi-square or Fisher’s exact tests. For non-categorical variables, a one-way independent analysis of variance was used, followed by post hoc analysis. SARS-CoV-2 nAb titers were expressed as geometric mean titers (GMTs), calculated using log-transformed individual titers and then reported as back-transformed titers. A generalized linear model was employed to test the significance of epidemiological factors such as age, gender, vaccination schedules, and intervals between immunization and blood sampling associated with the GMTs of nAb titers. The prevention of COVID-19 was evaluated using a Cox regression model to assess the effect of immunization schedule, age, and gender. Data analyses were performed using SAS software version 9.4 (SAS Institute, Inc., Cary, NC, USA), with a significance level set at *p* < 0.05.

## 3. Results

From 23 March to 20 July 2021, a total of 710 HCWs who received their first vaccination against COVID-19 were enrolled in this study. Of these participants, 493 (69.4%) were female, and the mean age was 42.1 ± 11.6 years (ranging from 20.0 to 79.5 years). The second and third vaccine doses were administered between June 2021 and April 2022, before the emergence of the Omicron variant epidemic in Taiwan in May 2022 (Figure 1). Figure 2 provides detailed information on the number of subjects on different immunization schedules. In the end, 668 (94.1%) out of the initial 710 participants completed three doses of prime-boost vaccine administration, and 290 (40.8%) participants received the fourth immunization dose during the Omicron epidemic. 

### 3.1. Reactogenicity of SARS-CoV-2 Vaccines of Different Platforms

The frequencies of both local and systemic adverse events (AEs) following vaccination with doses 1–4 within 7 days are depicted in Figure 3a–d and Appendix A. The AstraZeneca vaccine was exclusively used in the initial two doses, which along with the Moderna vaccine, was linked to a significant incidence of various AEs at moderate to severe levels (grade 2 to 3, Appendix A). It is noteworthy that in comparison to the Moderna vaccine, the rates of systemic AEs after AstraZeneca vaccination were notably higher for dose 1 but lower for dose 2 (Figure 3a,b). In this study, the Medigen protein-based subunit vaccine was introduced as the booster dose (dose 3) and additional dose (dose 4) of immunization. The rates of most local and systemic AEs following immunization with the Medigen subunit vaccine were significantly lower than those of the Moderna and Pfizer-BioNTech vaccines produced with the mRNA platform (Figure 3c). A similar trend was observed for the Novavax protein-based vaccine after dose 4 immunization, although the difference in AEs was not statistically significant, likely due to the relatively small number of subjects who received the additional dose (dose 4) of the vaccine (Figure 3d, Appendix A). The non-solicited AEs are listed in Appendix A. No serious AEs, including myocarditis, pericarditis, thrombosis, and thrombocytopenia, were identified in the vaccine recipients in this study. 

### 3.2. Immunogenicity of SARS-CoV-2 Vaccines and Dynamics of Vaccine-Induced Neutralizing Antibody (nAb) against SARS-CoV-2

The first three doses of vaccines were administered to the HCWs before the Omicron epidemic in Taiwan (Figure 1). This indicated that the measured antibody titers were purely evoked by immunization but not interfered with by natural infection. The GMTs of nAb against the ancestral strain of SARS-CoV-2 in subjects on different vaccination schedules of doses 1–3 are shown in Figure 4a,b and Appendix A. Before dose 3, the GMT titers were universally higher in the recipients of the Moderna vaccine than in those who received the AstraZeneca vaccine (Figure 4b). However, the antibody titers waned very quickly in all recipients regardless of the received vaccine type. Of note, the GMTs reached a level (9.8 IU/mL, 95% confidence limits [CLs], 9.00–10.68 IU/mL) approaching the pre-vaccination level before the third dose of vaccination in subjects who had received two homogeneous doses of AstraZeneca vaccines. The third dose vaccines, irrespective of vaccine types, significantly boosted the nAb titers, with the GMTs being greater in two mRNA vaccine recipients than in the protein-based Medigen vaccine recipients (Figure 4b). Nevertheless, the nAb titer waned again after dose 3. Among 290 subjects who had the fourth dose of vaccine, the pre-vaccination GMT declined to a level of 242.6 IU/mL (95% CLs, 213.4–275.8 IU/mL). The GMT of nAb was again boosted by the dose 4 vaccines, with the highest GMT for the new generation Moderna BA1 vaccine, followed by Moderna, Pfizer-BioNTech, Novavax and Medigen (Figure 4c, Appendix A).

### 3.3. Factors Associated with the GMTs of Vaccine-Evoked nAb against SARS-CoV2

To more comprehensively investigate the factors associated with the GMTs of nAb in the vaccinees after dose 2 and dose 3, a generalized linear model was used to evaluate the roles of interval after vaccination and subjects’ demographics in the immunogenicity of vaccine recipients on different vaccine schedules (Table 1). As expected, after dose 2, the GMTs of nAb were significantly associated with vaccine schedules, with the highest GMTs in recipients having the Moderna–Moderna homologous schedule. The subjects of female gender and younger age were also significant factors associated with higher GMTs of nAb after dose 2. After dose 3, the recipients of three doses of the Moderna vaccine had significantly higher GMTs than the other subjects on the other seven vaccination schedules. It was interesting to learn that the type of vaccine administered at the first dose could affect the post-third dose GMTs. For instance, the subjects on AstraZeneca followed by two doses of Moderna (AMM group, 740.4 IU/mL, 95% CLs, 699–784.3 IU/mL) had significantly lower nAb titers than those on three doses of Moderna (MMM group, 915.3 IU/mL, 95% CLs, 879.9–952.1 IU/mL, *p* < 0.0001, Appendix A). The gender and age factors were both not of statistical significance in this analysis. The interval after the first vaccination was not a significant factor of GMTs after dose 2 and dose 3.

### 3.4. Effect of SARS-CoV-2 Vaccine Schedules against COVID-19 during the Omicron Epidemic

Until late June 2023, COVID-19 was reported in 335 (51.0%) out of 710 participants through a questionnaire, review of medical records, and measurement of anti-N antibody (Appendix A). After excluding 53 and 5 participants with missing data on COVID-19 diagnosis and the primary three doses of the vaccination schedule, respectively, as well as 35 additional participants who received the fourth dose of the SARS-CoV-2 vaccine after contracting COVID-19, an analysis was carried out on the remaining 617 (86.9%) participants to study the impact of different vaccination schedules on preventing COVID-19.

The diagnosis of COVID-19 was documented in 229 (48.5%) of 617 participants. The diagnosis was not associated with the age or gender factors of the subjects (Table 2). Participants who received three doses of the Moderna vaccine at doses 1–3 seemed to have a lower risk of contracting COVID-19 compared to those who received the other eight vaccination schedules, although this difference was not statistically significant (*p* > 0.05 for all, Table 2). However, the administration of the fourth vaccine dose was significantly associated with a lower risk of contracting COVID-19, with a hazard ratio of 0.359 (*p* < 0.0001, Figure 5a). Among those who had received the fourth dose, the use of the new generation vaccine (Moderna BA1) was associated with the lowest risk of COVID-19 diagnosis, with a hazard ratio of 0.353 (95% confidence interval, 0.126–0.986, *p* = 0.0470) when compared to those who received the Pfizer-BioNTech vaccine (Figure 5b, Appendix A). 

## 4. Discussion

This study outlines key findings on the safety profiles and immunogenicity of various SARS-CoV-2 vaccines in HCWs in Taiwan. The results further offer crucial insights into the differing protective effects of various vaccine platforms and schedules.

The observed variations in AEs across different vaccine platforms highlight the importance of assessing reactogenicity to facilitate informed vaccine choices. The AstraZeneca adenoviral vector and Moderna mRNA vaccines were associated with higher frequencies of both local and systemic AEs, particularly after the first dose. These findings are consistent with earlier reports indicating higher reactogenicity profiles for these vaccine types [4,5]. Interestingly, while systemic AEs were more frequent following the first dose of AstraZeneca, they were lower after the second dose, suggesting a potential adaptive response to initial exposure. 

The Medigen protein-based vaccine demonstrates significantly lower reactogenicity compared to mRNA vaccines. This reduced incidence of adverse events, especially systemic reactions, positions the Medigen vaccine as a highly tolerable option, potentially increasing acceptance among populations sensitive to vaccine reactogenicity. The Novavax protein-based vaccine also follows a similar trend, although the differences are not statistically significant, possibly due to the smaller sample size for the fourth dose recipients. Importantly, our study found no serious adverse events, including thrombotic or cardiovascular events, for any of the vaccine types, firmly supporting the overall safety of these vaccines.

The robust antibody responses elicited by the initial vaccine doses corroborate the effectiveness of these vaccines in inducing an immune response. The Moderna mRNA vaccine consistently yielded higher GMTs of nAb than the AstraZeneca vaccine, consistent with the existing literature on the superior immunogenicity of mRNA platforms [19]. However, the rapid decline in antibody titers post-second dose across all vaccine types underscores the necessity for booster doses to maintain immunity.

The third dose significantly boosted nAb titers across all vaccine types, with the highest responses observed in recipients of the mRNA vaccines, followed by protein-based vaccines. This boost indicates the importance of heterologous (mix-and-match) schedules in sustaining higher antibody levels, as evidenced by previous studies [20,21]. Nevertheless, the waning of antibody titers post-third dose suggests that periodic boosters might be necessary for long-lasting immunity, particularly amidst emerging variants.

The fourth dose, administered during the Omicron outbreak, further amplified nAb titers, with the highest GMTs observed in recipients of the Moderna BA1 vaccine. This finding aligns with data supporting the enhanced immunogenicity of updated vaccines targeting specific variants [22]. The observed immunogenicity hierarchy among different vaccine types and doses spotlights the potential benefit of updated vaccine formulations in responding to evolving variants.

A major finding of this study was that the fourth vaccine dose significantly reduced COVID-19 incidence among HCWs. This outcome aligns with global data on the enhanced protection conferred by additional booster doses against Omicron and other variants [16]. Notably, the Moderna BA1 vaccine demonstrated the lowest risk of COVID-19, underscoring the efficacy of variant-specific booster formulations in improving pandemic control. Despite the differences in vaccine-induced antibody responses, the data did not show a statistically significant variance in COVID-19 infection rates among the primary three-dose schedules, suggesting that other immune mechanisms may contribute to vaccine efficacy, such as T-cell responses. This aligns with studies indicating that T-cell immunity is crucial in long-term protection against severe outcomes [23].

Our generalized linear model analysis identified several factors influencing nAb titers, including vaccine type, schedule, age, and gender. Younger age and female gender were associated with higher GMTs, particularly after the second dose, reflecting biological differences in immune response [24]. However, these factors were less significant post-third dose, perhaps due to the overriding effect of the additional booster. Notably, the type of vaccine administered initially appeared to influence subsequent antibody responses, with the AstraZeneca–Moderna combination yielding lower titers compared to the all-Moderna regimen. These findings highlight the importance of considering initial vaccine type and demographic factors in optimizing immunization strategies.

### Study Limitations

Several limitations should be considered. First, the study’s observational design might introduce selection bias, particularly in vaccine choice, which was left to participant preference. Second, the relatively small sample size for certain vaccine groups, especially those receiving the fourth dose, may limit the generalizability of the findings. Thirdly, we did not measure nAb using the live virus. Therefore, the nAb titers obtained through the ELISA method in this study should not be compared directly to those obtained through standard methods using live viruses. Additionally, it is important to note that the ELISA method measures the nAb titer against the original strain of SARS-CoV-2 and not against other variants. Additionally, while the study accounted for nAb titers, other immune responses, including T-cell immunity, were not assessed, potentially underestimating the vaccines’ protective effects. Finally, real-world adherence to the immunization schedule varied, potentially confounding the observed outcomes.

## 5. Conclusions

This study underscores the critical role of booster doses in maintaining and enhancing immunity against SARS-CoV-2, particularly with emerging variants like Omicron. Both mRNA and protein-based vaccines effectively boost nAb titers, with variant-specific mRNA vaccines like Moderna BA1 showing superior efficacy. The differential reactogenicity profiles among vaccine types underscore the need for tailored vaccination strategies to maximize uptake. Our findings support periodic booster vaccinations as a robust public health strategy to mitigate COVID-19 spread and protect vulnerable populations. Further research should focus on long-term immunity and the role of cellular responses in optimizing vaccine schedules and formulations.

## Figures and Tables

**Figure 1 vaccines-12-01057-f001:**
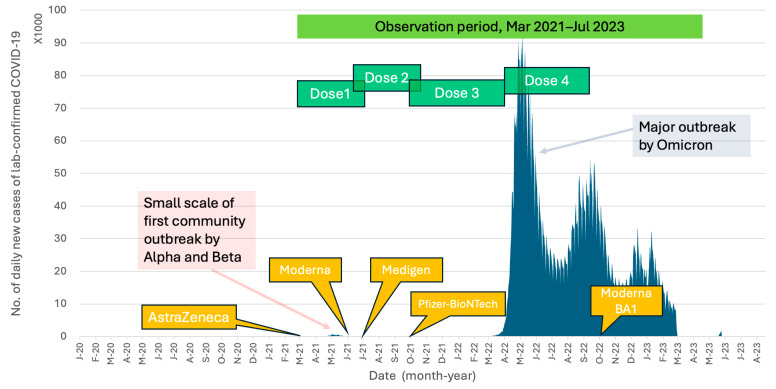
Timelines of the COVID-19 outbreaks and the introduction of various COVID-19 vaccines in Taiwan. The study period and the temporal relationship of vaccine doses administered to the healthcare workers in the study are also shown. The data of daily COVID-19 confirmed cases in Taiwan are available at https://covid-19.nchc.org.tw/2023_city_confirmed.php (accessed on 18 August 2024).

**Figure 2 vaccines-12-01057-f002:**
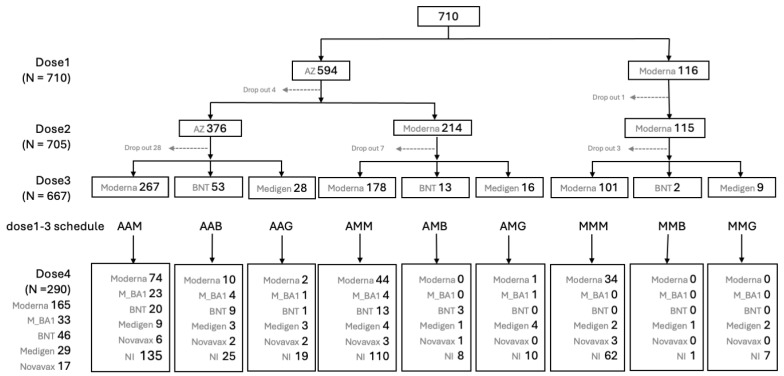
Flow chart of case enrollment and numbers of subjects on different vaccines at dose 1, 2, 3 and 4 against COVID-19.

**Figure 3 vaccines-12-01057-f003:**
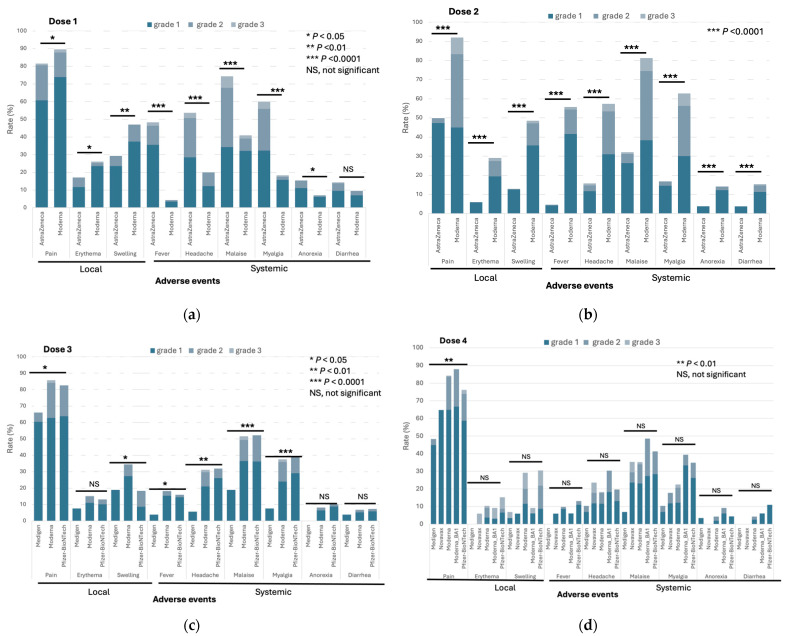
The rates of solicited adverse events of immunization against COVID-19 within seven days of doses 1 (**a**), 2 (**b**), 3 (**c**), and 4 (**d**).

**Figure 4 vaccines-12-01057-f004:**
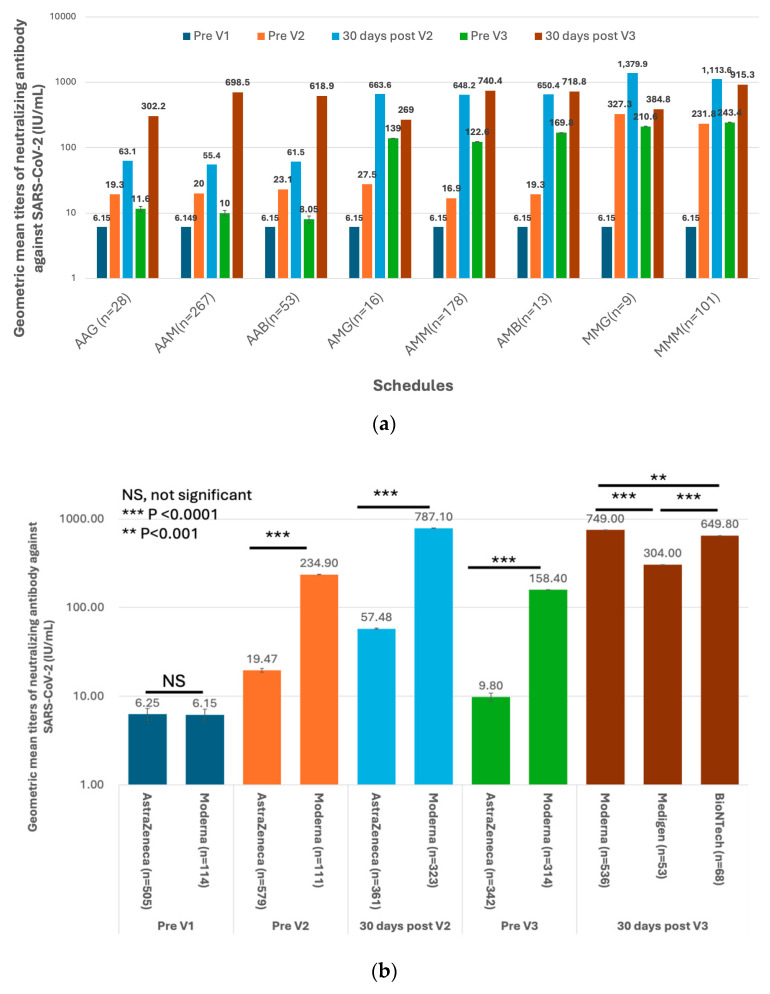
Humoral immunogenicity of vaccines against COVID-19 in healthcare workers. (**a**) The geometric mean titers (GMTs) of neutralizing antibodies (nAb) against the SARS-CoV-2 ancestral strain in eight groups of healthcare workers on different vaccination schedules on five occasions, including before vaccination (Pre V1), before the 2nd dose of vaccination (Pre V2), 30 days after the 2nd dose of vaccination (30 days post V2), before the 3rd dose of vaccination (Pre V3), and 30 days post the 3rd vaccination (30 days post V3). (**b**) The comparisons of GMTs evoked by different vaccines on five occasions. (**c**) The nAb before the fourth dose of vaccine (Pre V4) and 30 days after dose 4 (30 days post V4). The comparisons of post-immunization GMTs were made between Moderna vaccine recipients and those on the other four vaccines. Categorical variables were compared using the Chi-square or Fisher’s exact tests. For non-categorical variables, a one-way independent analysis of variance was used, followed by post hoc analysis.

**Figure 5 vaccines-12-01057-f005:**
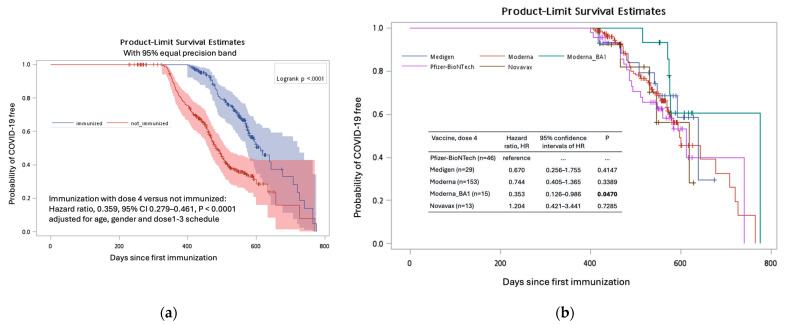
The Kaplan–Meier curves depict the probabilities of individuals remaining free from COVID-19 following their initial immunization against SARS-CoV-2. Subgroups were formed based on the administration of a fourth vaccine dose (immunized versus not immunized) (**a**) and the specific type of the vaccines among the dose 4 vaccine recipients (**b**).

**Table 1 vaccines-12-01057-t001:** Analysis of factors associated with the GMTs of nAb against SARS-CoV-2 after the 2nd and 3rd dose of vaccination in healthcare workers on different vaccine schedules.

Parameter	30 Days after Second Immunization			30 Days after Third Immunization		
	Estimate	Standard Error	*p* Value	Estimate	Standard Error	*p* Value
Dose 1–3 Vaccination Schedule $, vs. MMM						
AAB	−3.0117	0.3166	<0.0001	−0.4251	0.0822	<0.0001
AAG	−2.9765	0.2779	<0.0001	−1.1476	0.0981	<0.0001
AAM	−3.0824	0.1861	<0.0001	−0.2947	0.0656	<0.0001
AMB	−0.7316	0.4072	0.0729	−0.2889	0.1417	0.0028
AMG	−0.6025	0.3893	0.1222	−1.2444	0.1348	<0.0001
AMM	−0.6973	0.2803	0.0131	−0.2472	0.0977	<0.0001
MMG	0.15169	0.3835	0.6926	−0.9047	0.1333	<0.0001
Interval since V1	0.00404	0.0057	0.4793	0.0012	0.0008	0.1383
Gender female vs. male	0.28022	0.0947	0.0032	0.0620	0.0331	0.0615
Age, in years	−0.0077	0.0038	0.0452	−0.0011	0.0013	0.4196

$ The three letters of vaccination schedules indicate the respective manufacturers of COVID-19 vaccines administered at the 1st, 2nd, and 3rd immunization. Abbreviations: A, AstraZeneca; B, Pfizer-BioNTech; G, Medigen; M, Moderna; nAb, neutralizing antibody. A generalized linear model was used to evaluate the significance of epidemiological factors in relation to the GMTs of nAb titers. The subjects on Moderna–Moderna–Pfizer-BioNTech (MMB) were excluded from the analysis due to the small number of cases (N = 2).

**Table 2 vaccines-12-01057-t002:** Cox regression analysis of factors associated with the absence of COVID-19 diagnosis in 617 healthcare workers.

Parameter	Negative for COVID-19 N= 318	Positive for COVID-19 N = 299	Hazard Ratio (HR)	95% HR Confidence Limits	*p*
Female gender (%)	220 (69.2)	218 (72.9)	1.160	0.893–1.507	0.2674
Age, years, mean ± standard deviation	41.5 ± 11.3	42.7 ± 11.5	1.004	0.994–1.015	0.4464
Dose 4, vs. not immunized					
Not immunized	157 (49.4)	205 (68.6)	…	…	…
Medigen	20 (6.29)	9 (3.01)	0.307	0.153–0.614	0.0008
Moderna	98 (30.8)	55 (18.4)	0.372	0.275–0.505	<0.0001
Moderna_BA1	9 (2.83)	6 (2.01)	0.191	0.077–0.473	0.0003
Novavax	8 (2.52)	5 (1.67)	0.447	0.182–1.100	0.0796
Pfizer-BioNTech	26 (8.18)	19 (6.35)	0.409	0.250–0.668	0.0004
^$^ Dose 1–3, vs. MMM					
MMM	56 (17.6)	43 (14.4)	…	…	…
AAB	25 (7.86)	23 (7.69)	1.407	0.830–2.387	0.2051
AAG	11 (3.46)	16 (5.35)	1.386	0.772–2.486	0.2740
AAM	117 (36.8)	117 (39.1)	1.362	0.950–1.953	0.0924
AMB	6 (1.89)	7 (2.34)	1.497	0.659–3.402	0.3350
AMG	8 (2.52)	8 (2.68)	1.408	0.649–3.052	0.3863
AMM	88 (27.7)	80 (26.8)	1.262	0.863–1.845	0.2304
MMB	0 (0)	2 (0.67)	2.596	0.612–11.016	0.1957
MMG	6 (1.89)	3 (1.00)	1.160	0.356–3.788	0.8052

^$^ The three letters of vaccination schedules indicate the respective manufacturers of COVID-19 vaccines administered at the 1st, 2nd, and 3rd immunization. Abbreviations: A, AstraZeneca; B, Pfizer-BioNTech; G, Medigen; M, Moderna; nAb, neutralizing antibody. A Cox regression model was used to evaluate the hazard ratios of demographics of participants and various immunization schedules for the prevention of COVID-19. The subjects on Moderna–Moderna–Pfizer-BioNTech (MMB) were excluded from the analysis due to the small number of cases (N = 2).

## Data Availability

The raw data of the immunogenicity and safety profile post-immunization against COVID-19 in the participants can be downloaded from https://1drv.ms/f/s!Aoh-bMMfe01Qhst1zyBzXTlbIcehEg?e=IWOscq (accessed on 18 August 2024).

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
