# Peer review of "Evaluation of Vaccine Strategies among Healthcare Workers during COVID-19 Omicron Outbreak in Taiwan"

_vaccines, 2024, doi:10.3390/vaccines12091057_

Round 1

Reviewer 1 Report

Comments and Suggestions for Authors

This is an interesting study on the effect of vaccination against COVID-19 in preventing new infections and producing anti-SARS-CoV-2 antibodies in Taiwan. Although the effectiveness of vaccines against COVID-19 have been documented by a plethora of studies, the report is well written and it is worth publishing. A few comments to be addressed by the authors:

(a) Rates in AE refer to 100,000 vaccination doses? It is not clearly depicted in the charts or in the text. Please clarify. 

(b) The gold standard to identify neutralization bodies is to perform a neutralization assay with live virus. The authors have not peformed such an assay and therefore the measurement of antibodies in serum is an indirect way. Moreover, the assay should be performed against different strains. This is a drawback of the study and it should be mentioned

(c) Where are any differences in probability of contracting COVID-19 between the immunization groups?  (i.e. any differences betwen those who chose a mixed vaccination scheme against those who were vaccinated with the same type of vaccine in all doses?). This would be also interesting to be depicted.

(d) Where there any major AE such as VITT in patients receiving AZ?

Comments on the Quality of English Language

The language is fine minor editing is needed. 

Reviewer 2 Report

Comments and Suggestions for Authors

REVIEWER'S REPORTManucsript title: Evaluation of vaccine strategies among healthcare workers during COVID-19 Omicron outbreak in Taiwan (Authors: Lin et al.).

  This manuscript covers the statistical analysis of the immunogenicity and reactogenicity of three types SARS-CoV-2 vaccines, as well as a comparison of how effectively they protected healthcare workers (HCWs) from COVID-19 during Taiwan's Omicron outbreak. The observational trial, which ran from March 15, 2021 to July 15, 2023, included healthy HCWs who had not previously received COVID-19 immunisation. The vaccines offered to the participants (HCWs) were protein-based (Medigen, Novavax), mRNA 17 (Moderna, BioNTech-Pfizer), and adenovirus-vectored (AstraZeneca).  In my opinion, the statistical assessments are fairly robust, and the manuscript is reasonably well described, and it is well suitable for publishing in this journal. Nonetheless, I believe that before to publishing, some minor changes and/or supplements are needed just in a few places throughout the manuscript's text, specifically:

In subsection Statistical Analysis (pages 3–4), it would benefit the readers if the methods of statistical evaluation were depicted in the Scheme in addition to being described in the text. More references should be provided in this subsection.

 In Materials and Methods, the sentences in lines 96-99 could be somewhat modified, by writing " The study's participants were healthy healthcare workers who had not received the SARS-CoV-2 vaccine and had no intention of receiving immunosuppressive or immune-modifying medication. They should not have gotten a COVID-19 vaccination or participated in any investigational study"

 Overall, the manuscript's material is written in quite decent, understandable English; only a few minor corrections are needed.

Comments on the Quality of English Language

Reasonably clear and understandable English. There are only a few text spots that require minor modification.
